# Frailty Severity and Cognitive Impairment Associated with Dietary Diversity in Older Adults in Taiwan

**DOI:** 10.3390/nu13020418

**Published:** 2021-01-28

**Authors:** Wei-Ching Huang, Yi-Chen Huang, Meei-Shyuan Lee, Hsing-Yi Chang, Jia-Yau Doong

**Affiliations:** 1Graduate Institute of Life Sciences, National Defense Medical Center, Taipei 11490, Taiwan; a0975386905@gmail.com (W.-C.H.); hsingyi@nhri.org.tw (H.-Y.C.); 2Institute of Population Health Sciences, National Health Research Institute, Miaoli 35053, Taiwan; 3Department of Nutrition, China Medical University, Taichung 40402, Taiwan; yichenhuang@mail.cmu.edu.tw; 4School of Public Health, National Defense Medical Center, Taipei 11490, Taiwan; mmsl@mail.ndmctsgh.edu.tw; 5Department of Nutritional Science, Fu Jen Catholic University, New Taipei City 24205, Taiwan

**Keywords:** cognitive, frailty, cognitive frailty, dietary diversity, Nutrition and Health Survey in Taiwan (NAHSIT)

## Abstract

This study aims to investigate whether frailty severity in conjunction with cognitive function, termed as” cognitive frailty”, is associated with dietary diversity in older adults. This cross-sectional study used the data from the 2014–2016 Nutrition and Health Survey in Taiwan (*N* = 1115; age ≥ 65 years). Dietary intake was assessed using a 24 h dietary recall and food-frequency questionnaire, and dietary diversity score (DDS; range, 0–6) and food intake frequency were calculated. The presence of frailty phenotypes was determined using the FRAIL scale, which was proposed by the International Association of Nutrition and Aging, and cognitive function was assessed using the Mini–Mental State Examination (MMSE) score. The prevalence of cognitive frailty (FRAIL scale score ≥ 3 and MMSE score ≤ 26) was 4.2%. A higher consumption frequency of dairy products, whole grains, vegetables, fruit, fish and seafood, nuts, tea, and coffee, as well as lower pickled vegetable, was inversely associated with cognitive frailty. Those with prefrailty or frailty and lower DDS demonstrated a higher cognitive impairment risk (adjust odds ratio (OR) = 2.15, 95% confidence interval = 1.21–3.83) than those without frailty and higher DDS. Older adults with cognitive prefrailty or cognitive frailty were associated with lower DDS, and frailty with lower DDS was associated with worsening cognitive function.

## 1. Introduction

It is suggested that frailty interacts with cognitive impairment in the decline cycle of ageing [1], whose interaction results in adverse health outcomes such as malnutrition [2], poor quality of life, disability, hospitalization, and death [3,4]. The International Academy of Nutrition and Aging (IANA) and the International Association of Gerontology and Geriatrics (IAGG) proposed the concept and definition of “cognitive frailty” as follows: “a presence of frailty (or prefrailty) and cognitive impairment, excluding Alzheimer’s disease or other dementia [5]”. The result of a meta-analysis also demonstrated that frailty is predictable of cognitive impairment and dementia incidence [6]. As a result, the early combined detection and prevention of frailty and cognitive decline may be essential.

Eating habits, are a modifiable factor for healthy aging [7]. Inadequate dietary intake and multiple nutrient deficiencies or a monotonous diet may be associated with frailty and cognitive impairment [8,9]. Dietary diversity score (DDS) is a simple measurement of dietary quality that can predict all-cause and cause-specific mortality [7] as well as medical expenditure in older adults in Taiwan [10]. Better DDS in older adults was an indicator of nutrient adequacy and health status [7], which might enhance survival related to cognitive impairment [4]. Some dietary patterns, such as plant-based diets, and the Mediterranean dietary pattern are known to be associated with cognitive maintenance [11,12]. Moreover, the association between different food types and the severity of frailty combined cognitive function remains unknown. The purpose of this study is two fold: first, to examine whether frailty severity in conjunction with cognitive function, termed as “cognitive frailty”, using the IANA definition, was associated with dietary diversity in older adults; second, to investigate whether frailty and lower DDS in older adults were associated with a higher risk of cognitive impairment.

## 2. Methods

### 2.1. Study Participants

This cross-sectional study used representative data from the 2014–2016 Nutrition and Health Survey in Taiwan (NAHSIT). The data of all 1440 community-dwelling older adults (age ≥65 years) were included. And outlined in detail in other sources [13,14], the NAHSIT’s design and sampling method involved data collection through face-to-face household interviews and physical examinations. Among all the NAHSIT respondents, 1156 older adults completed the Mini–Mental State Examination (MMSE) and FRAIL scale questionnaires. Both the data of two study participants who had a medical history of dementia and those of 39 participants with aberrant total daily energy intakes (outside the ranges of 800–4200 kcal/day for men and 500–3500 kcal/day for women) were excluded [15]. With these criteria, data on 1115 older adults were eligible for analysis. All participants provided informed consent forms, and this study was approved by the Ethics Committees of Academic Sinica, Taiwan before being conducted.

### 2.2. Data Source

During the NAHSIT household interviews, sociodemographic characteristics, medical histories, and responses on a 24 h dietary recall questionnaire, a 79-item simplified food-frequency questionnaire (SFFQ), a 36-item Short-Form Health Survey (SF-36) [16], and the MMSE [17] were collected.

#### 2.2.1. Frailty

Frailty assessment was based on the five-item FRAIL scale proposed by the IANA [18]; the items are fatigue, resistance, ambulation, illnesses, and weight loss. One point was allocated to each condition under each item. Fatigue was measured by asking the participants the following question: “How much time during the past 4 weeks did you feel tired?” The responses included “all the time” and “most of the time”. Resistance and ambulation were assessed based on the difficulty the study participants encountered when “climbing one flight of stairs” and “walking one block”, respectively. Illness was assessed based by participants reporting five or more illnesses of the 30 illnesses included in the FRAIL scale. Weight loss was assessed according to weight loss over time. The total overall scores ranged from 0 to 5, —with scores of 3–5 indicating frailty, 1–2 indicating prefrailty, and 0 indicating robustness. The FRAIL scale has been validated in previous studies [18,19].

#### 2.2.2. Cognitive Function Assessment

Cognitive function was assessed using the MMSE, which includes seven items: orientation, registration, attention, calculation, recall, language, and visuospatial construction. The MMSE scores range from 0 to 30 (the higher the score, the better the cognition), which has been extensively used in clinical studies in Chinese ethnicity, and its validation has been published elsewhere [20]. In this study, scores >26 and ≤26 defined normal and impaired cognition, respectively [21].

#### 2.2.3. Cognitive Frailty

In this study, cognitive frailty and cognitive prefrailty were defined as the concomitant presence of an MMSE score of ≤26 and a FRAIL scale scoring ≥3 and 1–2, respectively.

#### 2.2.4. Dietary Intake Information

Dietary intake information was obtained using the simplified food frequency questionnaire (SFFQ) and a 24 h dietary recall. The SFFQ includes 79 food items classified into 21 food categories, and participants were asked how often they had consumed each food item per month, week, and day over the previous month. In this study, we calculated the daily consumption frequency of given food item. The similar SFFQ designed for the Elderly Nutrition and Health Survey in Taiwan has been validated in a previous study [22].

Dietary quality was assessed based on DDS, obtained from food intake information by a 24 h dietary recall [23]. According to the Taiwanese Food Guides, dietary diversity was assessed for six food groups: fruit, vegetables, grains, meat, dairy, and oil/fat/seeds/nuts. The ‘meat group’ comprised protein rich foods, i.e., soybean products eggs, fish, shellfish, and meats. For each of the six food groups, a score of 1 was given if more than half a serving per day was consumed; the scores ranged from 0 to 6 [7].

### 2.3. Statistical Analysis

All data were weighted to represent the older population in Taiwan during 2014–2016. SAS (version 9.4, SAS Institute, Cary, NC, USA) was used for analysis, and SUDAAN (version 11.0.3, Research Triangle Park, NC, USA) was used to account for the survey sampling effect.

Analysis of variance (ANOVA) was used to analyze the differences in the distributions of continuous variables. For significant ANOVA results, the Bonferroni multiple-comparison test was used as a post hoc test. Categorical variables were compared using Chi-square tests.

To evaluate the association between frailty severity and cognitive function, a multiple logistic regression model was employed. In addition, statistical power was limited for frail older adults, where we have merged the categories of prefrailty and frailty as ‘frailty status’. We considered age (65–69, 70–74, 75–79, and ≥80 years), sex, residential area, education level (illiterate, up to primary school, and high school and above), perceived health status (good, fair, and poor), DDS (>4 and ≤4), and body mass index (BMI; <18.5, 18.5–23.9, 24–26.9, and ≥27 kg/m^2^) as the potential covariates. Frailty severity and DDS were also considered to predict cognitive function.

To ensure that missing BMI did not create bias in the results for frailty severity and cognitive impairment, a sensitivity analysis was conducted on five datasets with multiple imputations [24]. A two-tailed *p*-value < 0.05 indicated statistical significance.

## 3. Results

Table 1 presents the baseline characteristics of the older NAHSIT participants by frailty severity, in which 402 (37.3%) and 64 (6.2%) older adults exhibited prefrailty and frailty, respectively. The older adults with frailty were older and exhibited lower physical activities, apart from having a lower MMSE score, and poorer perceived health. We also did sex-specific analyses, and found that frail older women had a higher BMI (23.9 ± 0.29, 25.1 ± 0.52, and 27.6 ± 1.32, sorted by the severity of frailty, *p* < 0.05), and a higher rate of obesity (BMI ≥ 27, *p* < 0.05), but the data analysis did not show a similar result to frail older men. (data not shown). Moreover, a higher proportion of older adults with frailty had diabetes mellitus, hypertension, kidney disease, and heart disease, whereas a higher proportion of those with prefrailty had hyperlipidemia.

Table 2 presents the distribution of DDS, food intake, and blood biochemistry factors by frailty severity and cognitive function. The prevalence of cognitive frailty was 4.2%, and that of prefrailty with cognitive impairment was 20.8%; both groups demonstrated significant lower DDS compared with the healthy group. Those with cognitive impairment had lower energy intake regardless severity of frailty. Frail older adults with normal cognition had lower daily protein intakes, but not cognitive frailty. Those with cognitive frailty had significantly lower intake frequencies for dairy products, whole grain, fruit, meat, nuts and seeds, tea, coffee, and vegetables, along with lower energy intakes, compared with the healthy group. The intake frequency of pickled vegetables was significantly higher among those with cognitive prefrailty, but significantly lower among those with frailty and no cognitive impairment. Furthermore, those with cognitive frailty had significantly lower hemoglobin levels, but significantly higher hemoglobin A1c, triacylglycerol, and blood urea nitrogen levels (*p* < 0.05).

Table 3 presents the association between frailty severity and cognitive status. In the crude model, older adults with prefrailty and frailty had 2.0- and 3.5-fold, respectively, higher odds of cognitive impairment than the healthy group. Model 1 adjusted for age, sex, and sampling stratum. Model 2 additionally adjusted for education, DDS, and BMI, which was then additionally adjusted for smoking and perceived general health to obtain Model 3. In Model 3, the odds ratios (ORs; 95% confidence intervals (CIs)) for prefrailty and frailty were 1.56 (1.12–2.18) and 2.23 (0.75–6.68), respectively. Those with prefrailty had a 56% higher risk of cognitive impairment than the healthy group (OR, 1.56; 95% CI, 1.12–2.18). However, these differences were non-significant for those with frailty (OR, 2.23; 95% CI, 0.75–6.68). Because some participants had missing BMI data, sensitivity analysis was performed on five datasets generated using multiple imputation; the sensitivity analysis yielded consistent results.

Frailty status combined with DDS (≤4 and >4) in relation to cognitive function is illustrated in Figure 1. The DDS–cognitive function interaction was significant (*p* = 0.048). Older adults with frailty and an unvaried diet demonstrated a higher risk of cognitive impairment (OR, 2.15; 95% CI, 1.21–3.83) than those without frailty and a varied diet.

## 4. Discussion

In this study, the prevalence of cognitive frailty and prefrailty was 4.2% and 20.8%, respectively—consistent with the study that reported a low cognitive frailty prevalence (1.0%–12.1%) among community-dwelling older adults [25]. Older adults with cognitive frailty exhibited a lower proportion of better DDS (DDS > 4) and lower consumption frequency for dairy products, whole grains, vegetables, fruit, meat, nuts, tea, and coffee. In addition, those with frailty and normal cognition had a significantly higher consumption of dairy products and significantly lower consumption of whole grains, tea, and pickled vegetables.

Older adults with a high BMI had a higher frailty risk [26]. Through the analysis by sex, we found that obesity (BMI ≥ 27 and high waist circumference) was associated with a higher prevalence of frailty in women, but not in men. Our observation is consistent with a French study [27]. In that study, Monteil D et al. reported that frailty is more common in obese non-institutionalized women, but not men. It is also known that obesity is associated with oxidative stress and low-grade systemic inflammation and neuroinflammation, which may be associated with frailty [28]. This is further suggested by a Finnish longitudinal study. This was a study of 1119 people aged 30 years or older without frailty at baseline. After more than 22 years of follow-up, the results indicated that obesity, which may begin during the middle age, is an underlying cause of frailty [26]. The national, representative Taiwanese study demonstrated that older adults with BMI 24–26.9 kg/m^2^ had the lowest mortality among the older population. As such, weight reduction among older people is not recommended [29]. A study has shown that oxidative stress also contributes to higher frailty in addition to BMI [30]. Frailty and prefrailty are associated with higher inflammatory parameters, such as C-reactive protein and interleukin 6 [31]. In more detail, dietary patterns with more omega-3 fatty acid, vegetables, fruits, nuts, coffee, and tea have been found as active components linking anti-inflammatory and antioxidative [32], and both components have been shown to be inversely associated with frailty [13].

Consumption of a single food group or supplement alone did not alleviate frailty and cognitive impairment [28]. Suggested by a few studies, dietary diversity and lifestyle components were inversely associated with cognitive decline [9], frailty [8], and mortality [7]. Particularly, the Chinese cohort study on 4749 older adults aged more than 80 years also used SFFQ and MMSE to measure food intake and cognitive function, respectively, whose results demonstrated that, in comparison with those consuming them rarely or never older adults consuming vegetables, fruit, meat, and soy products daily exhibited a significant reduction in cognitive decline and mortality risks [9].

Further looking into Asian diets, pickled vegetables have often been studied as a specific dietary pattern, but its association with cognitive function in older adults has been proved inconsistent. In a Taiwanese study on older adults, the “traditional dietary pattern” (characterized by a high pickled vegetable intake) protected against the decline in logical memory recall [33]. By contrast, in a Chinese longitudinal study involving 4847 participants aged 55 years or older, it was approved that the “starch-rich dietary pattern”, in particular salted vegetables and legumes, was positively associated with cognitive decline [34]. In the current study, high pickled vegetable intake was positively associated with cognitive impairment and frailty.

It has been shown that protein-rich food intakes were inversely associated with cognitive decline and frailty [13], but the clinical trial study showed that protein intakes exceeding 0.8 g/kg/d did not increase lean body mass, muscle performance, physical function, or well-being in older men with physical function limitations. Likewise, whether this association with protein-rich food applied to a frail older adults was still unclear [35]. This is inconsistent with our results that a higher intake of protein-rich food such as dairy products, fish, and other seafood is associated with a lower prevalence of cognitive frailty.

In our study, we found that those with prefrailty had a 56% higher risk of cognitive impairment than those from the healthy group with a relatively lower risk of cognitive impairment. Older adults with a lower DDS in conjunction with either prefrailty or frailty had a 2.15-fold higher risk of cognitive impairment than those without frailty or with a higher DDS (Figure 1). It is evident that the consumption of foods rich in antioxidants and polysaccharides may reduce oxidative damage and protect against oxidative stress, thus delaying neurodegenerative decline (both cognitive and motor); such foods include vegetables, fruit, cereals, beans, mushrooms, tea, milk products, and meat [32]. Corresponding to this finding, the older adults with a higher DDS in our study demonstrated a higher consumption frequency of vegetables, fruit, dairy products, and mushrooms (Appendix A).

The strength of this study is that the data used were representative of community-dwelling Taiwanese older adults. The data also included comprehensive sociodemographic characteristics as well as dietary and nutrition-related biomarker information. With these, our result is informational for the nutrition-related health care system that aims to mitigate frailty severity and improve cognitive function.

Nonetheless, this study has several limitations. First, this study was cross-sectional, which would hamper causal inference. Subsequently, prospective cohort studies should have been conducted. Second, DDS was calculated from 24 h dietary recall questionnaires, which may not reflect long-term dietary habits, even though the dietary pattern of older adults tended to be unchanged [36], as confirmed by the SFFQ data. Lastly, dietary information was collected using both the SFFQ and 24 h dietary recall questionnaires, of which the reliability and validity of dietary intakes may be impacted by memory problem, memory loss, or cognitive decline.

## 5. Conclusions

In conclusion, a higher DDS and consumption frequency of dairy products, whole grains, vegetables, fruit, fish and other seafood, nuts, tea, and coffee, as well as lower pickled vegetable consumption frequency, were inversely associated with cognitive frailty. In addition, older adults with frail status combined with lower DDS were demonstrated to be associated with worsening cognitive function. Accordingly, further research is necessary to confirm whether dietary interventions could reduce cognitive frailty in older adults in Taiwan.

## Figures and Tables

**Figure 1 nutrients-13-00418-f001:**
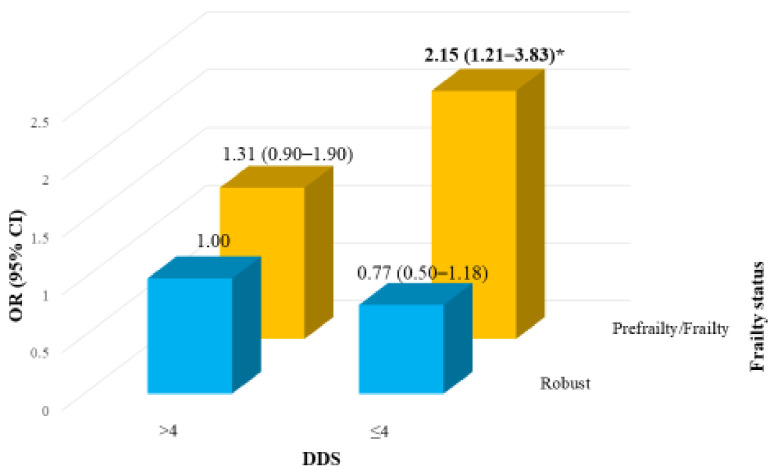
Joint odds ratios for cognitive impairment in frailty severity by dietary diversity score (DDS) (adjusted for age, sex, sampling stratum, education, body mass index, smoking, and general health); * *p* = 0.048 for interaction. OR, odds ratio; CI, confidence interval.

**Table 1 nutrients-13-00418-t001:** Characteristics of older adults by frailty severity in the 2014–2016 Nutrition and Health Survey in Taiwan (*N* = 1115) *.

Characteristic	Frailty Severity			*p* ^†^
	Robust	Prefrailty	Frailty	
*n* (%)	649 (56.6)	402 (37.3)	64 (6.2)	
Men, %	47.8	48.6	33.5	0.126
Age (years), %	72.4 ± 0.3	75.3 ± 0.5	78.2 ± 1.4	<0.001
65–69	40.1	23.0	12.0	<0.001
70–74	26.1	23.7	16.7	
75–79	19.6	24.0	27.6	
≥80	14.3	29.3	43.7	
Education, %				0.145
Illiterate	7.7	12.7	14.4	
Up to primary school	44.5	47.3	48.9	
High school and above	47.9	39.5	36.7	
Smoking, %	26.7	34.1	25.1	0.084
Alcohol use, %	48.0	46.4	47.2	0.932
Physical activity (METs/week)	24.7 ± 1.5	21.7 ± 1.4	12.5 ± 1.9	<0.001
MMSE (score)	26.2 ± 0.3	25.0 ± 0.3	23.0 ± 0.9	<0.001
BMI (kg/m^2^), %	24.2 ± 0.2	25.1 ± 0.4	26.7 ± 0.9	0.034
<18.5	3.5	3.6	0.0	0.037
18.5–23.9	48.0	36.2	24.5	
24–26.9	31.4	31.8	28.4	
≥27	17.1	28.4	47.1	
Perceived economic status, %				0.212
More than enough	15.1	14.8	18.5	
Just enough	58.8	53.9	43.4	
Some difficulties	20.8	20.7	26.3	
Very difficult	5.3	10.6	11.9	
Perceived health status, %				<0.001
Good	36.9	27.7	19.5	
Fair	52.8	45.3	23.8	
Poor	10.3	27.0	56.8	
Diease history, %				
Diabetes mellitus	13.8	27.3	53.0	<0.001
Hypertension	44.6	59.6	72.9	<0.001
Hyperlipidemia	11.6	25.0	19.3	<0.001
Kidney disease	1.5	5.5	9.3	0.006
Heart disease	8.5	22.6	29.9	<0.001

Abbreviations: BMI: body mass index; MMSE: Mini–Mental State Examination. * All data were weighted for the unequal probability of being sampled in the SUDAAN; categorical variables are presented as percentages and continuous variables as means ± standard errors. ^†^ Continuous and categorical variables were analyzed using analysis of variance and chi-square tests, respectively.

**Table 2 nutrients-13-00418-t002:** Comparison of dietary diversity score, selected food intake, and nutritional biomarkers with frailty severity and cognitive function in older adults in the 2014–2016 Nutrition and Health Survey in Taiwan.

Cognitive Function	Frailty Severity					
	Robust		Prefrailty		Frailty	
	Normal	Impaired	Normal	Impaired	Normal	Impaired
*n* (%)	385 (35.3)	264 (21.3)	179 (16.5)	223 (20.8)	17 (2.0)	47 (4.2)
DDS, %	4.93 ± 0.06	4.77 ± 0.07	4.95 ± 0.06	4.69 ± 0.08 *	5.17 ± 0.29	4.62 ± 0.21
>4	73.1	65.1	81.9	64.4	73.2	54.5
≤4	26.9	34.9	18.1	35.6	26.8	45.5
Energy, kcal/day	1854 ± 46.8	1589 ± 47.0 *	1907 ± 66.0	1602 ± 67.5 *	1867 ± 173	1426 ± 92.5 *
Daily nutrient densities (/1000 Kcal)						
Carbohydrate, g/day	139 ± 2.22	151 ± 2.22 *	137 ± 2.52	138 ± 2.71	141 ± 7.43	139 ± 3.65
Fat, g/day	30.7 ± 0.79	26.7 ±0.84 *	32.2 ± 0.91	31.2 ± 1.06	32.9 ± 2.55	32.1 ± 1.48
Protein, g/day	42.7 ± 0.74	39.2 ±0.84 *	41.2 ± 1.03	42.0 ±1.06	37.9 ± 1.95 *	41.5 ± 2.27
Food intake frequency, times/day						
Dairy products	0.53 ± 0.05	0.38 ± 0.04 *	0.49 ± 0.09	0.42 ± 0.04	1.49 ± 0.33 *	0.24 ± 0.11 *
Whole grains	0.51 ± 0.05	0.34 ± 0.06 *	0.53 ± 0.09	0.35 ± 0.07	0.24 ± 0.10 *	0.21 ± 0.12 *
Vegetables	3.60 ± 0.18	3.19 ± 0.18	3.20 ± 0.14 *	2.75 ± 0.15 *	3.42 ± 0.44	2.91 ± 0.43
Pickled vegetable	0.10 ± 0.01	0.12 ± 0.02	0.10 ± 0.02	0.17 ± 0.03 *	0.03 ± 0.02 *	0.16 ± 0.08
Fruit	1.52 ± 0.08	0.97 ± 0.06 *	1.32 ± 0.08	1.04 ± 0.08 *	1.72 ± 0.48	0.96 ± 0.15 *
Soybean	0.48 ± 0.04	0.42 ± 0.04	0.51 ± 0.04	0.44 ± 0.06	0.59 ± 0.24	0.44 ± 0.08
Fish/seafood	1.22 ± 0.07	0.91 ± 0.07 *	1.10 ± 0.10	0.93 ± 0.08 *	0.90 ± 0.21	1.47 ± 0.17
Egg	0.40 ± 0.02	0.34 ± 0.03 *	0.47 ± 0.09	0.36 ± 0.03	0.30 ± 0.06	0.39 ± 0.06
Livestock	0.57 ± 0.03	0.55 ± 0.07	0.57 ± 0.05	0.55 ± 0.04	0.45 ± 0.09	0.39 ± 0.07 *
Poultry	0.20 ± 0.01	0.18 ± 0.02	0.20 ± 0.02	0.19 ± 0.02	0.31 ± 0.10	0.16 ± 0.02
Processed meat	0.13 ± 0.02	0.08 ± 0.01	0.16 ± 0.02	0.13 ± 0.02	0.19 ± 0.05	0.20 ± 0.04
Nuts and seeds	0.38 ± 0.04	0.21 ± 0.03 *	0.32 ± 0.05	0.18 ± 0.03 *	0.46 ± 0.16	0.20 ± 0.09 *
Tea	0.44 ± 0.04	0.39 ± 0.06	0.49 ± 0.06	0.28 ± 0.05 *	0.12 ± 0.09 *	0.18 ± 0.09 *
Coffee	0.34 ± 0.04	0.19 ± 0.04 *	0.30 ± 0.05	0.21 ± 0.06	0.27 ± 0.16	0.10 ± 0.05 *
Snacks	0.45 ± 0.04	0.33 ± 0.05 *	0.53 ± 0.08	0.38 ± 0.06	0.63 ± 0.17	0.32 ± 0.08
Fried food	0.03 ± 0.01	0.01 ± 0.00 *	0.04 ± 0.01	0.02 ± 0.00	0.02 ± 0.01	0.03 ± 0.01
Sweet beverage	0.36 ± 0.04	0.35 ± 0.05	0.33 ± 0.04	0.38 ± 0.06	0.21 ± 0.11	0.27 ± 0.08
Nutritional-related blood biochemistry (n = 650)						
TC, mg/dL	193 ± 3.55	186 ± 3.76	189 ± 4.25	179 ± 4.24 *	189 ± 16.68	187 ± 9.68
Hb, g/dL	13.6 ± 0.11	13.2 ± 0.17 *	13.7 ± 0.24	12.7 ± 0.23 *	12.2 ± 0.50 *	12.7 ± 0.32 *
HbA1C, %	6.02 ± 0.08	6.27 ± 0.08 *	6.07 ± 0.09	6.44 ± 0.23	6.20 ± 0.31	7.02 ± 0.28 *
LDL, mg/dL	122 ± 3.29	118 ± 3.11	119 ± 4.11	111 ± 4.39	117 ± 13.16	115 ± 9.85
HDL, mg/dL	55.5 ± 1.74	52.0 ± 1.31	54.1 ± 2.14	51.6 ± 1.38	52.2 ± 6.04	49.7 ± 5.17
TG, mg/dL	125 ± 6.33	118.8 ± 6.18	128 ± 9.49	118 ± 7.62	139 ± 27.7	160 ± 20.3
BUN, mg/dL	16.1 ± 0.47	17.0 ± 0.59	17.4 ± 0.50 *	18.5 ± 0.89 *	21.0 ± 2.87	20.9 ± 1.97 *
CRE, mg/dL	0.83 ± 0.02	0.89 ± 0.04	0.92 ± 0.04	0.97 ± 0.06 *	1.10 ± 0.14	1.06 ± 0.14

Abbreviations: DDS: dietary diversity score; TC: total cholesterol; Hb: hemoglobin; HbA1C: Hemoglobin A1c; LDL: low-density lipoprotein; HDL: high-density lipoprotein TG: triacylglycerol, BUN: blood urea nitrogen, CRE: creatinine. All data were weighted for the unequal probability of being sampled in the SUDAAN; categorical variables are presented as percentages and continuous variables as means ± standard errors. *p* values were for the one-way analysis of variance and using Bonferroni multiple-comparison and chi-square tests for categorical variables were analyzed using analysis of variance: significant differences were relative to the normal group (normal cognition and without frailty). ** p* < 0.05.

**Table 3 nutrients-13-00418-t003:** Cognitive impairment risks with frailty severity in older adults by logistic regression in the 2014–2016 Nutrition and Health Survey in Taiwan.

	Frailty Severity			*p*-Value
	Robust	Prefrailty	Frailty	
Cognitive impairment/normal	264/385	223/179	47/17	
Crude	1.00	2.09 (1.59˗2.75)	3.55 (1.57˗8.03)	<0.001
Model 1	1.00	1.70 (1.21˗2.40)	2.36 (1.05˗5.30)	0.005
Mode 2	1.00	1.71 (1.18˗2.47)	2.58 (0.87˗7.71)	0.011
Model 3	1.00	1.56 (1.12˗2.18)	2.23 (0.75˗6.68)	0.020

Model 1: adjusted for age, sex, and sampling stratum; Model 2: Model 1 + education, dietary diversity score, and body mass index; Model 3: Model 2 + smoking and general health.

## Data Availability

Data analyzed in this paper (article) were collected by the research project “Nutrition and Health Survey in Taiwan (NAHSIT2014-2016)” sponsored by the Department of Health in Taiwan (MOHW110-HPA-H-114-144703). All participants provided informed consent forms, and this study was approved by the Ethics Committees of Academic Sinica, Taiwan, before being conducted.

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
