# Peer review of "Frailty Severity and Cognitive Impairment Associated with Dietary Diversity in Older Adults in Taiwan"

_nutrients, 2021, doi:10.3390/nu13020418_

Round 1

Reviewer 1 Report

The submitted manuscript aims at assessing the possible relationship between cognitive frailty and DDS using a large national database and a cross-sectional design. The results suggest that cognitive frailty and pre-frailty are associated with lower DDS. Also, frailty/pre-frailty and lower DDS are associated with cognitive impairment. Although the aim of the study is relevant, the study design and the analysis of the database have limitations which limit the relevance and the interest of the results.

  1. The results show that frailty in the population studied is associated to higher BMI. This evidence should be better addressed by the authors. Monteil D et al reported (J Nutr Health Aging 2020) that frailty is more common in obese non institutionalized women, but not men. It is therefore of key imprtance that the authors analyze their data considering the gender.
  2. The authors discuss the relationship between obesity and frailty by suggesting that oxidative stress and inflammation are increased in obese older adults which in turn may favour frailty. Although scientifically sounding, this hypothesis is in contrast with recent evidence obtained in a different population (i.e., Australian citizens). Tembo MC et al. (Am J Mens Health 2020) showed that there is a relationship between oxidative stress and frailty, independently from BMI. Therefore, higher BMI may not be the sole explanation for the higher incidence of frailty.
  3. The cross sectional design of the study does not allow to ascertain whether frailty/cognitive impairment is the cause or the consequence of poor dietary habit. This limitation of the study, correctly outlined by the authors, may result in minimal clinical relevance of the results. From the discussion, it may appear that the authors suggest that dietary diversity could be involved in the pathogenesis of cognitive impairment. However, this appear as a speculation, at least based on the results obtained, since the authors do not report any quantitative information on the daily caloric and protein intake.
  4. Qualitative dietary information were obtained based on a validated food-frequency questionnaire. It is widely acknowledged that reliability of food-frequency questionnaire are influenced by the cognitive status of the individuals tested. Therefore, it could be speculated that the poor dietary habits of cognitively impaired older adults are secondary to unreliable recall of food eaten rather by an evidence in nature. The authors should address how the eating behavior of cognitively impaired older adults were validated.

Minor points

Table 2, "Impaied": please, correct

Author Response

  1. The results show that frailty in the population studied is associated to higher BMI. This evidence should be better addressed by the authors. Monteil D et al reported (J Nutr Health Aging 2020) that frailty is more common in obese non institutionalized women, but not men. It is therefore of key imprtance that the authors analyze their data considering the gender.

 Response: Thank you for this suggestion. We did sex specific analyses, and found that frail older women had higher BMI and obesity rate, but not men, as shown in Table below.

Table. Body composition measurement of older adults in the 2014–2016 Nutrition and Health Survey in Taiwan*

Characteristic

Frailty

p

Robust

Prefrailty

Frailty

Men

n (%)

335 (57.3)

206 (38.4)

28 (4.4)

BMI (kg/m2), %

24.5±0.33

25.1±0.45

25.2±0.93

0.641

< 18.5

1.69

1.98

0.00

0.445

18.5-23.9

43.5

35.0

35.1

24-26.9

33.5

39.1

25.4

≥ 27

21.4

23.9

39.5

Waist circumference

89.9±0.74

91.9±1.29

96.1±3.42

0.145

Hip circumference

93.5±0.57

94.3±1.08

93.2±1.43

0.976

Women

n (%)

314 (56.0)

196 (36.3)

36 (7.8)

BMI (kg/m2), %

23.9±0.29

25.1±0.52

27.6±1.32

0.009

< 18.5

5.00

5.18

0.00

0.008

18.5-23.9

52.0

37.4

17.5

24-26.9

29.7

24.8

30.4

≥ 27

13.4

32.7

52.1

Waist circumference

82.5±0.71

87.2±0.96

95.6±2.64

<0.001

Hip circumference

93.2±0.57

95.9±0.88

101±1.94

0.001

*All data were weighted for the unequal probability of being sampled in the SUDAAN; categorical variables are presented as percentages and continuous variables as means ± standard errors.

†Continuous and categorical variables were analyzed using analysis of variance and chi-square tests, respectively.

According to the Tables above, we revised the manuscript in Lines 129-131 and 195-197 accordingly.

  1. 2. The authors discuss the relationship between obesity and frailty by suggesting that oxidative stress and inflammation are increased in obese older adults which in turn may favour frailty. Although scientifically sounding, this hypothesis is in contrast with recent evidence obtained in a different population (i.e., Australian citizens). Tembo MC et al. (Am J Mens Health 2020) showed that there is a relationship between oxidative stress and frailty, independently from BMI. Therefore, higher BMI may not be the sole explanation for the higher incidence of frailty.

 Response:  We are most grateful to the reviewer for making this point. Higher BMI may not be the sole explanation for the higher incidence of frailty, especially in men.

We have made revision in lines 204-209: In addition, higher BMI may not be the only explanation for the higher incidence of frailty in men. Frailty and prefrailty are associated with higher inflammatory parameters, such as C-reactive protein and Interleukin 6. In more details, dietary pattern with more omega-3 fatty acid, vegetables, fruits, nuts, coffee, and tea have been found active components linking anti-inflammatory and antioxidative, and both components have been shown inversely associated with frailty.

  1. The cross sectional design of the study does not allow to ascertain whether frailty/cognitive impairment is the cause or the consequence of poor dietary habit. This limitation of the study, correctly outlined by the authors, may result in minimal clinical relevance of the results. From the discussion, it may appear that the authors suggest that dietary diversity could be involved in the pathogenesis of cognitive impairment. However, this appear as a speculation, at least based on the results obtained, since the authors do not report any quantitative information on the daily caloric and protein intake.

Response: We appreciate your comment. The result of a Japanese study shows that frail older adults did not had lower daily protein intakes Motokawa et al. (J Nutr Health Aging 2018), however the study did not consider cognitive function. Therefore, we add to analysis the daily of energy and macro-nutrients intake in Table 2, and revised in Lines 145-148:

  1. Qualitative dietary information were obtained based on a validated food-frequency questionnaire. It is widely acknowledged that reliability of food-frequency questionnaire are influenced by the cognitive status of the individuals tested. Therefore, it could be speculated that the poor dietary habits of cognitively impaired older adults are secondary to unreliable recall of food eaten rather by an evidence in nature. The authors should address how the eating behavior of cognitively impaired older adults were validated.

Response: Thank you for your concern. As a Co-PI in Nutrition and Health Survey in Taiwan (NAHSIT), I am very confident about the simplified FFQ data from NAHSIT which were 1) The NAHSIT’s design was excluding older people who had a medical history of dementia or lost communication ability. 2) This survey uses trained interviewers and face to face to conduct 24-hour dietary recall and FFQ and cross-interrogate with young people or primary caregivers in the family to confirm that the answer is reliable. 3) It is evident that FFQ is a reasonable method in older adults, even have cognitive impairment. Morris et al. (Am J Epidemiol 2003). Although we aimed to have an unbiased measurement, possible bias may still unavoidable. This is acknowledged as a limitation in Lines 249-251. Nevertheless, the impact in our findings might be limited.

  1. Minor points

Table 2, "Impaied": please, correct

Response: Thank you. Table 2 has corrected.

Reviewer 2 Report

Authors aimed to examine the relationship between frailty severity and cognitive function with dietary diversity in older adults, and examine whether frail older adults with higher dietary diversity had lower risk of cognitive impairment.

I do not understand the "Cognitive Frailty" score. It is a composite of the 2 other scores, one that assessed physical frailty and one that assessed cognitive function. I don't see the point of this third composite score.

When discussing frailty and BMI (line 188) it is worth mentioning the potential inverse relationship between BMI and poor health in the elderly, as a low BMI/weight loss can be indicative of poor outcomes in the elderly.

Line 214, Which is the better predictor?  It is unclear how DDS and frailty status compare in their ability to predict cognitive function. 

Author Response

  1. I do not understand the "Cognitive Frailty" score. It is a composite of the 2 other scores, one that assessed physical frailty and one that assessed cognitive function. I don't see the point of this third composite score.

Response: Thank you for your comment. Frailty and cognitive function impairment are two perilous clinical conditions for older adults. The International Academy of Nutrition and Aging (IANA) and the International Association of Gerontology and Geriatrics proposed the concept and definition of “cognitive frailty”. We aimed to examine whether frailty severity conjunction cognitive function (cognitive frailty) by using IANA definition is associated with dietary diversity in older adults.

A meta-analysis also demonstrated that frailty can predict cognitive impairment and dementia incidence, so our second aimed to investigate whether frail older adults with lower DDS were associated with a higher risk of cognitive impairment. To make it more clearly, we revised in lines 53-55.

  1. When discussing frailty and BMI (line 188) it is worth mentioning the potential inverse relationship between BMI and poor health in the elderly, as a low BMI/weight loss can be indicative of poor outcomes in the elderly.

Response: Thank you for the suggestion. We have now made this point in Lines: 202-204.

  1. Line 214, Which is the better predictor? It is unclear how DDS and frailty status compare in their ability to predict cognitive function.

Response: We analyzed the associated between DDS and cognitive function, and found that lower DDS were associated with a higher risk of cognitive impairment (data not shown). In this study, frailty and DDS were not associated with cognitive function individually. (p = .048 for interaction, in Figure 1.) In addition, dietary interventions are one of the major non-pharmacological approaches that are effective in improving frailty and cognitive decline in older adults. Hsieh et al. (Int J Behav Nutr Phys Act 2019); Valls-Pedret et al. (JAMA Intern Med 2015).

We also revised in lines 231-234: In our study, we found that those with prefrailty had a 56% higher risk of cognitive impairment than did the healthy group.  Older adults with a lower DDS in conjunction with either prefrailty or frailty had a 2.15-fold higher risk of cognitive impairment than did those without frailty or with a higher DDS (Figure 1).

Again, thank you for giving us the opportunity to strengthen our manuscript with your valuable comments and queries. We are hopeful that the Editors find this response adequate to allow a favorable decision about our paper.

Round 2

Reviewer 1 Report

The authors addressed the concerns raised by the first draft of their manuscript. No further concern from my side.

Author Response

thank you for giving us the opportunity to strengthen our manuscript with your valuable comments and queries.

Reviewer 2 Report

Grammar is still an issue as many sections are still incorrectly phrased and at times meaning is difficult to understand. For example, sentences 48-49 and 146-147 are unclear, to name two. There are many punctuation errors that affect clarity. Specific aims, I believe should state, "First, we aimed to examine
whether frailty severity in conjunction with cognitive function, termed "Cognitive Frailty" using IANA definition, is associated with dietary diversity in older adults. Second, we aimed to investigate whether frailty and lower DDS in older adults were associated with higher risk of cognitive impairment."

49-50 needs more explanation. All dietary patterns?  What does this mean?

Sentences 196-197 and 204-205 do not agree. This paragraph does not make a clear point.
